# Characteristics of Solar-Induced Chlorophyll Fluorescence in the Three River Headwaters Region, Qinghai-Tibetan Plateau during 2001 to 2020

**Jun Miao** [1,2], **Fei Xing** [3] and **Ru An** [1,*]

1   School of Earth Sciences and Engineering, Hohai University, 8 Focheng West Road, Nanjing 211100, China; 190209120008@hhu.edu.cn
2   School of Geography and Planning, Huaiyin Normal University, Huaian 223300, China
3   Department of Geography and Planning, University of Saskatchewan, 117 Science Place, Saskatoon, SK S7N 5C8, Canada; xingfei@hhu.edu.cn
*   Correspondence: anrunj@hhu.edu.cn; Tel.: +86-182-6000-2994

**Abstract:** The ecology of the Three River Headwaters Region (TRHR) is related to the long-term sustainable development of Qinghai Province and the whole of China. The change in chlorophyll fluorescence is an important index to measure the ecological environment. Therefore, it is of great significance to study the spatial and temporal distributions of Solar-induced chlorophyll fluorescence (SIF)and the related influencing factors in the TRHR. In this study, a high-resolution SIF dataset (2001–2020) was selected to be averaged on a time scale of years and months to investigate the annual and seasonal SIF characteristics, and the influencing climate factors were analyzed in combination with meteorological data by statistical method. The results showed that the SIF values ranged from 0.05 to 0.073 during 2001–2020, with a peak value of 0.073 in 2005 and 2009 and a minimum value of 0.05 in 2002. The averages of SIF values were higher in the source regions of the Yellow River source region (YR) and Langcang (Mekong) River source region (LCR) than in the source region of the Yangtze River source region (YZR). The SIF values of the TRHR in July, August and September were significantly higher than those in other months. The maximum value occurred in August at 0.11, and the minimum value was 0.008 in December. The precipitation had greater effect on the inter-annual variations in SIF. The monthly variation of SIF is influenced by precipitation, temperature and relative humidity. In addition, the influence of human activities and altitude on SIF should not be ignored. The results have certain reference value for protecting vegetation in the TRHR, and provide a reference for other regions to analyze the spatiotemporal changes and influencing factors by using SIF data.

**Keywords:** climate factors; remote sensing; solar-induced chlorophyll fluorescence; the Three River Headwaters Region





## 1. Introduction

As the main body of terrestrial ecosystems, vegetation plays an important role in connecting the soil, atmosphere and water systems [1]. Vegetation is also highly vulnerable to interference, damage, and other factors affecting climate and environmental changes, and the dynamic changes in vegetation often serve as important biological indicators of climate change, especially under the background of global climate change. Significant climate change has affected, and will continue to affect, the growth of vegetation, and the growth status of plants will also change to a certain extent [2–5]. Solar-induced chlorophyll fluorescence (SIF) is an excited-light phenomenon of the chlorophyll molecules in vegetation under natural lighting conditions. SIF occurs in the photoreaction process of vegetation photosynthesis and is closely related to the linear electron transfer rate [6,7]. Compared to traditional vegetation indices and spectral reflectance, SIF can more directly reflect the

true physiological state of vegetation and is considered an ideal "probe" for vegetation photosynthesis [8–10]. Moreover, accurate measurements of terrestrial carbon sinks are the key to achieving the goal of "dual carbon" (carbon peaking and carbon neutrality); however, uncertainties remain with regards to the intensities and locations of carbon sinks [11]. Gross primary production (GPP) is the amount of carbon fixed by vegetation photosynthesis and is also a key component in the global carbon cycle [12]. Flux observations are the most accurate way to estimate GPP but are limited by the number and distribution of sites [13]. Satellites have great advantages in assessing temporal and spatial changes in ecosystem GPP at the landscape, regional and global scales. Light energy utilization models have a clear principle, and their calculation is simple; thus, such models are widely used in GPP estimations, but the model parameters are usually estimated from vegetation indices, and this increases the uncertainty of the resulting GPP estimations [14]. SIF is considered as the 650–850-nm electromagnetic radiation emitted by chlorophyll molecules during photosynthesis under light, with two peaks of red light and far-red light occurring near 685 nm and 740 nm; the peak changes are closely related to the physiological state of vegetation [15]. Satellite SIF remote sensing is very sensitive to changes in the photosynthetic state of vegetation [16], showing great potential for estimating regional or global GPP [17]. Therefore, monitoring SIF changes is critical for understanding the photosynthetic status of vegetation and for studying the carbon cycle.

The development of remote sensing technology has made it possible to monitor SIF at large scales. Over the past ten years, SIF remote sensing has attracted extensive attention from scientists from all over the world. A growing body of research suggests that satellite-derived SIF products provide a new measure of global GPP. With the development of high-spectral-resolution satellite technologies, research on SIF satellite remote sensing inversions is increasing daily and mainly includes shared dataset products based on SCIAMACHY, GOME-2, GOME_F, OCO-2 and TROPOspheric Monitoring Instrument (TROPOMI) satellite data [18,19]. Existing SIF data have long been limited by low spatial resolutions and sparse data sampling. The TROPOMI sensor of Sentinel-5P launched in 2017 can significantly improve the spatial and temporal resolutions of SIF observations. However, the short temporal coverage of the data records limits their application in long-term research. With the deepening of SIF-application research in ecology and related fields, generating global, high-resolution, spatiotemporally continuous long-time-series SIF products has become critical. The SIF dataset (2001–2020) released by the National Qinghai-Tibet Plateau Science Data Center provides global, high-resolution SIF data covering a long time series from 2001 to 2020, thus effectively alleviating the above problems [20]. These products have been widely used in analyses of the spatiotemporal variations in, and influencing factors of, SIF.

The Three River Headwaters Region (TRHR) is among the areas with the highest concentration of high-elevation biodiversity in the world. It is also one of the areas with the most abundant water resources in the world, so it is known as the "Chinese Water Tower". The TRHR is the birthplace of the Yangtze River, the Yellow River and the Lancang River, serving as an important supply of freshwater resources in China; it is also a sensitive area and initiator of climate change in Asia, in the northern hemisphere and even across the world [21]. The Tibetan Plateau, on which the TRHR is located, is an extremely important part of the ecological security barrier of the Qinghai-Tibet Plateau in China. This plateau plays an important role in the construction of national ecological civilization, in promoting ethnic unity, and in maintaining the stability of Tibetan areas. It is related to national ecological security and the long-term development of the Chinese nation [22]. Therefore, the TRHR is a sensitive and ecologically fragile region under the context of climate change, so it is critical to study the dynamic changes in vegetation cover and the laws associated with climate impacts in the TRHR and to protect and restore the fragile and sensitive ecological environment under the background of climate change. However, no relevant research has explored the changes in SIF in the TRHR, let alone the long-term spatiotemporal variations or the related influencing climate factors.

Therefore, this study will analyze the spatio-temporal distribution and influencing factors of SIF in TRHR for the first time, and then how to carry out the spatio-temporal analysis of SIF and study the influencing factors, mainly from the following two aspects: first, SIF data with high precision were selected to analyze the inter-annual and inter-monthly SIF variation in TRHR; then, meteorological factors were selected to analyze the influencing factors of SIF change in TRHR. The paper is divided into the following sections: (1) introduction: mainly introducing the research background and significance of the paper; (2) materials and methods: introducing the research area, the data and methods used; (3) results: the inter-annual and inter-monthly variation in SIF and meteorological data in TRHR were analyzed. (4) discussions the influencing factors of the inter-annual and inter-monthly variation of SIF were analyzed by combining meteorological data; (5) conclusion: the results obtained in this paper are summarized.

## 2. Materials and Methods

### 2.1. Study Area

The TRHR is located in southern Qinghai Province, China. It is located between the 31°39′ and 36°12′ N latitude lines and the 89°45′ and 102°23′ E longitude lines, with a total area of 363,000 square kilometers and an average elevation of 3500–4800 m. It is considered as the roof of the world and the hinterland of the Qinghai-Tibet Plateau. The TRHR is one of the areas in China with the most concentrated distribution of glaciers. Snow-capped mountains, glaciers and rivers are widely distributed, and lakes and swamps are numerous; in addition, this region has the highest elevation, the largest area and the most abundant wetland across the world. The climate belongs to the Qinghai-Tibet Plateau climate system, a typical plateau continental climate characterized by alternating hot and cold seasons, distinct dry and wet seasons, small annual temperature differences, large daily temperature differences, long hours of sunshine, strong radiation, and a lack of four seasons. The study area was shown in Figure 1.

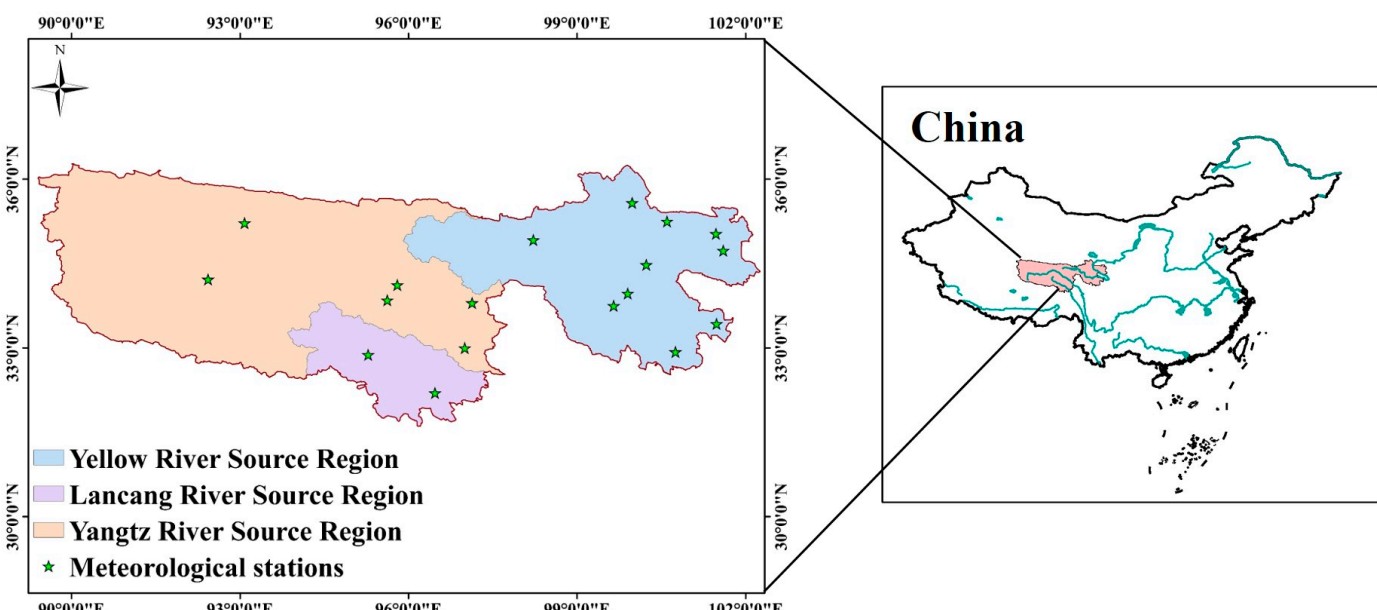

**Figure 1.** Study region of the TRHR and distribution of meteorological stations (MS).

### 2.2. SIF Data

The global, high-resolution SIF dataset (2001–2020) used herein was produced based on the XGBoost machine learning model. The main inputs were MODIS land surface reflectance data, land surface temperature and land type products, CERES reanalysis data, and C3/C4 vegetation cover data. TROPOMI SIF (RTSIF) data covering the period of 2001–2020 were reconstructed under clear skies. The dataset was validated against the

TROPOMI SIF and tower-based SIF data and compared with other satellite-derived SIFs (GOME-2 SIF and OCO-2 SIF), demonstrating the accuracy of the data. This dataset has high spatial and temporal resolutions (0.05° and 8 days, respectively) and is valuable for assessing photosynthesis and global carbon and water fluxes in terrestrial ecosystems over time, thus contributing to ecosystem carbon cycling and carbon neutrality studies [23]. The data can be downloaded from National Tibetan Plateau/Third Pole Environment Data Center for free (https://doi.org/10.6084/m9.figshare.19336346.v2 (accessed on 5 May 2023)). Based on the high accuracy and long time span of this data, RTSIF data is adopted in this paper. It is worth mentioning that the spatial resolution of SIF data used is 0.05°. Only lakes larger than 5.5 km × 5.5 km can be identified, but most lakes in this region are smaller than this [24]. Although there are many lakes in the TRHR region, this can be basically ignored on a large scale. ArcGIS was used to process SIF data to obtain the spatio-temporal distribution map of TRHR, and the mean statistics of the SIF data in different regions were carried out to obtain the line chart of SIF changes.

### 2.3. Meteorological Data

The meteorological data used in this study were obtained from the China Meteorological Data Network (http://data.cma.cn/site/index.html/ (accessed on 10 May 2023)) and included the data of the 18 meteorological stations in the TRHR from 2001 to 2020. The dataset included daily temperature, daily precipitation, daily average wind speed and daily relative humidity data. The downloaded weather data were averaged both monthly and annually; these time periods were consistent with the monthly and annual periods for which the average SIF data were obtained.

### 2.4. Model Application and Analysis

SIF data from 2001 to 2002 were extracted by vector boundary of TRHR. The annual and monthly variation data of SIF were obtained by averaging the data by year and month. In addition, the meteorological data are also averaged according to the inter-annual and inter-monthly variations. Pearson's correlation coefficient ($R^2$) and significant value (P) were used to analyze the influencing factors between SIF data and meteorological factors, and the level of $R^2$ reflected whether meteorological factors would have an impact on SIF.

## 3. Results

### 3.1. Annual SIF in the TRHR

The SIF data within one year is averaged to obtain the SIF value year by year. The annual mean SIF values were processed to obtain an average annual SIF distribution map of the TRHR. The results are shown in Figures 2 and 3. From 2001 to 2020, the overall SIF value of the TRHR ranged from 0.05 to 0.073, with a maximum value of 0.073 in 2005 and 2009, and a minimum value of 0.05 in 2002. The overall SIF value showed a gradual upward trend. The SIF value in TRHR shows spatial characteristics of high in the east and low in the west.

In order to further analyze the spatial difference, the TRHR was divided into the Yangtze River source region (YZR), Langcang (Mekong) River source region (LCR), and Yellow River source region (YR) for further analysis. The maximum and minimum SIF values in YZR were 0.053 and 0.025 respectively, appearing in 2005 and 2001. In the Southeast region of TRHR, the SIF value was higher than in other regions. The maximum and minimum SIF values in LCR were 0.111 and 0.050, respectively, appearing in 2005 and 2002, showing a spatial difference characteristic of being higher in the southeast and lower in the northwest. The highest and lowest SIF values in the source area of YR occurred in 2018 and 2001, which were 0.103 and 0.071, respectively, showing a trend of higher in the east and lower in the west. The SIF value change trends in these three regions were generally consistent. The SIF values in LCR and YR were significantly greater than those in YZR from 2001 to 2020. High-value areas were concentrated mainly in the eastern region of YR and the southern region of LCR.

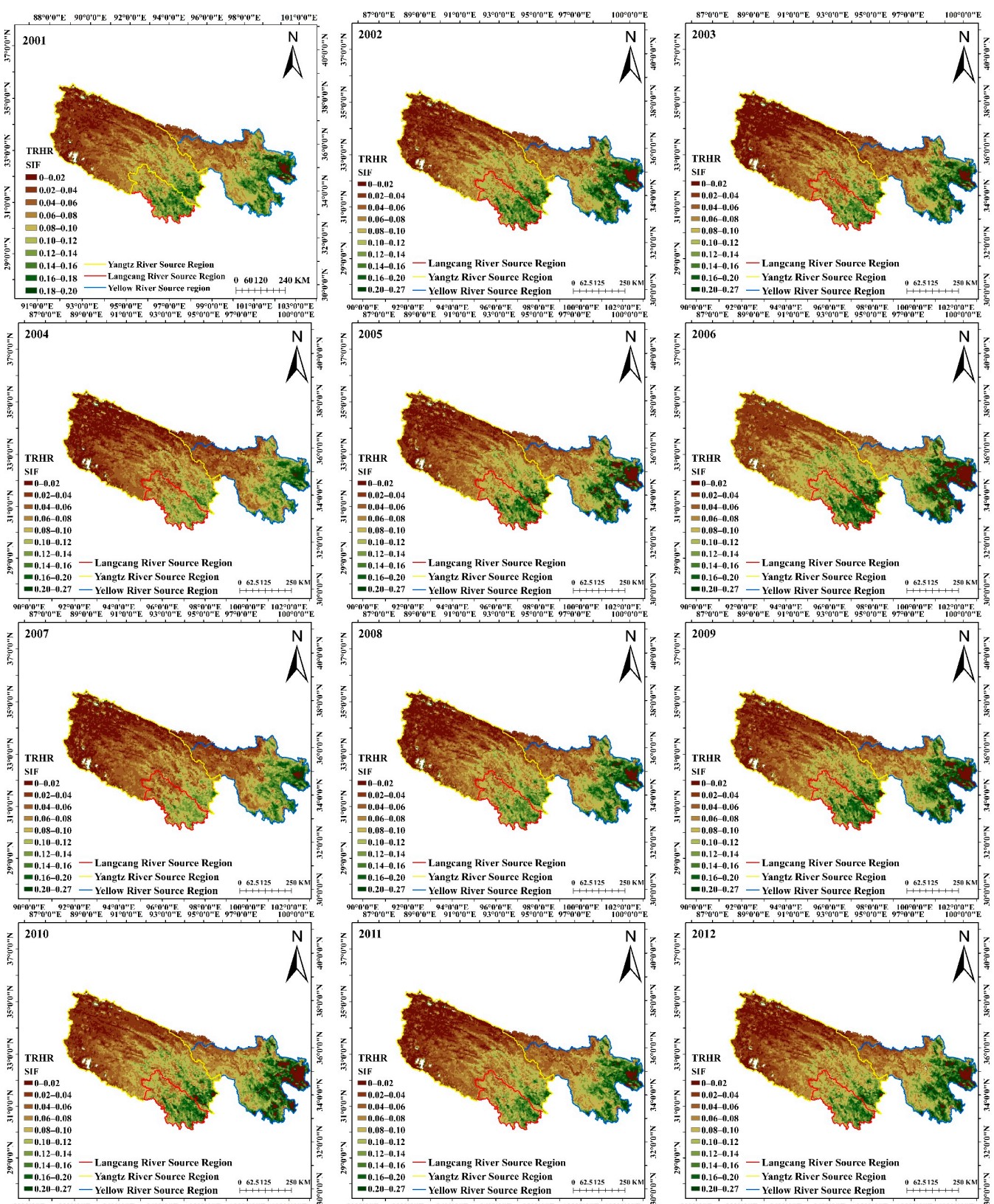

**Figure 2.** *Cont.*

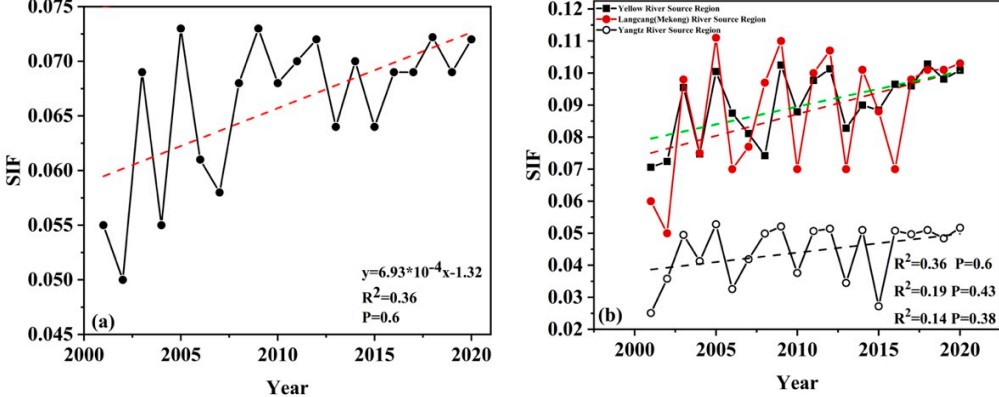

**Figure 2.** Spatial distribution of the annual SIF in the TRHR during the 2001–2020 period.

**Figure 3.** (**a**) Inter-annual variations in SIF in the TRHR; (**b**) inter-annual variations in SIF in YZR, LCR and YR.

### 3.2. Monthly SIF in the TRHR

The monthly average SIF value was obtained by averaging the SIF data of each month, and the results are shown in Figures 4 and 5. From January to May, the SIF value showed an upwards trend overall, reaching 0.055. It rose sharply from June, reached the maximum value of 0.107 in August, and then started to decrease and continued to decrease until December. The lowest SIF value occurred in December, at 0.008. Previous studies have shown that the beginning period for vegetation growth in the TRHR spans from days of the year 125–155, the end period of growth spans from days 280–290, and the length of the growing season lasts between 130 and 160 days. Thus, the SIF changes identified herein were consistent with the phenological periods in the TRHR [25]. From January to May and November to December, there was little difference in SIF values between the east and west of TRHR. From June to October, the SIF values showed a trend of high in the east and low in the west.

The maximum and minimum values in YZR appear in August and January, which were 0.0097 and 0.096, respectively. The maximum and minimum SIF values in LCR appear in August and December, which were 0.144 and 0.010, respectively. The maximum and minimum values of SIF in YR in a year were 0.147 and 0.007, in July and January. From January to May, the SIF values of the three regions did not differ extensively, and all showed a slowly increasing trend. From May, the SIF values of YR and LCR increased significantly more than those in YZR, and these increases continued until October. From November to December, the SIF values of the three areas were basically consistent. The SIF values of YR and the LCR did not differ much, but both were significantly higher than those of YZR.

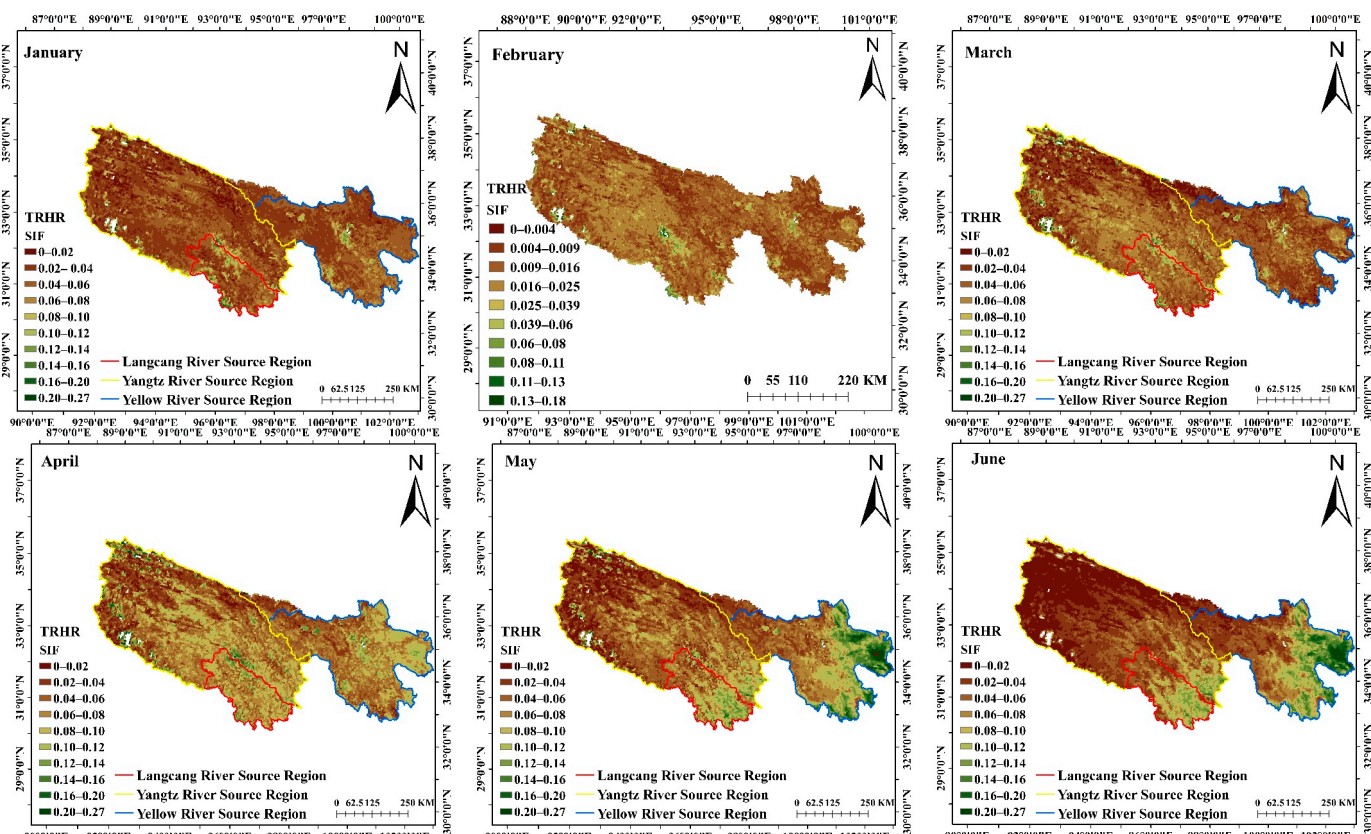

**Figure 4.** *Cont.*

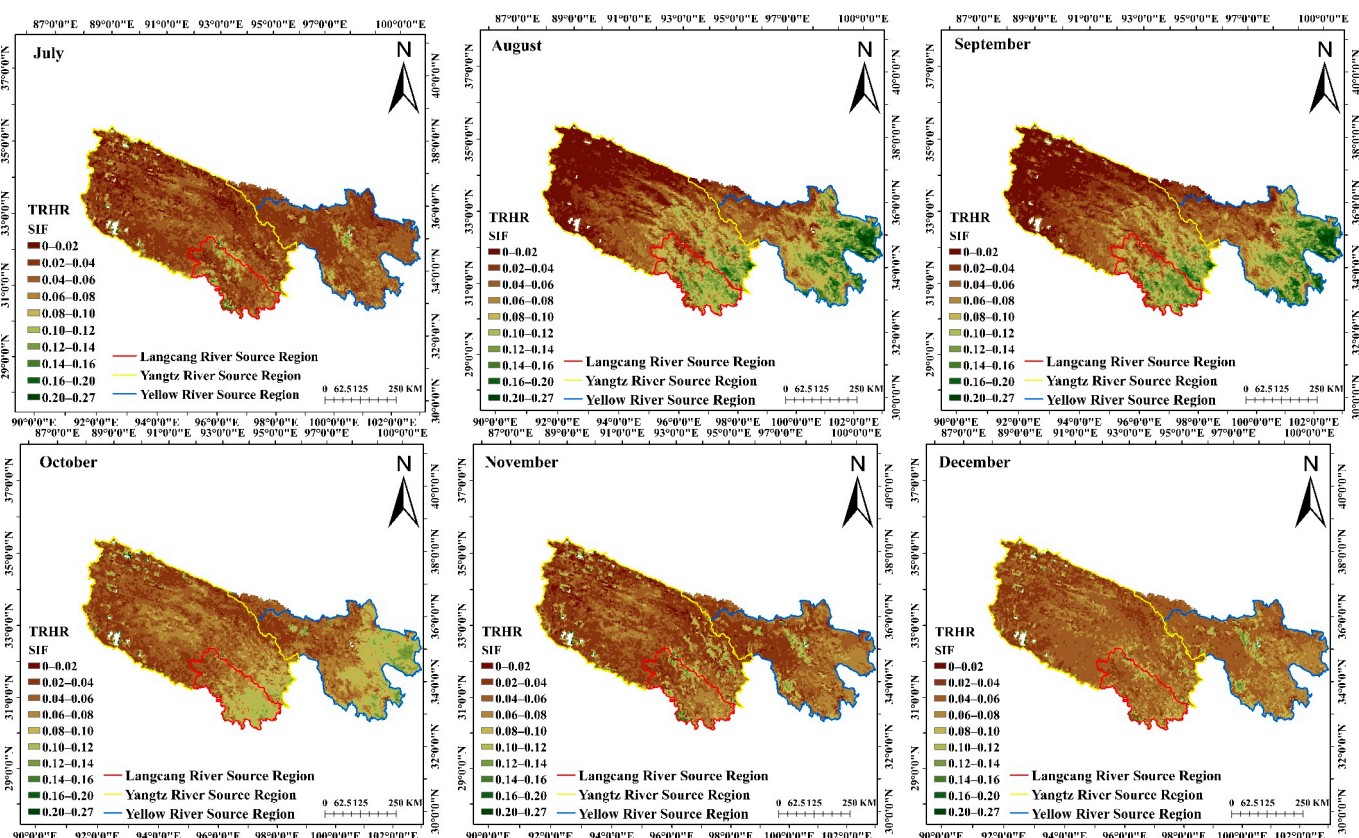

**Figure 4.** Spatial distribution of SIF in the TRHR from January to December.

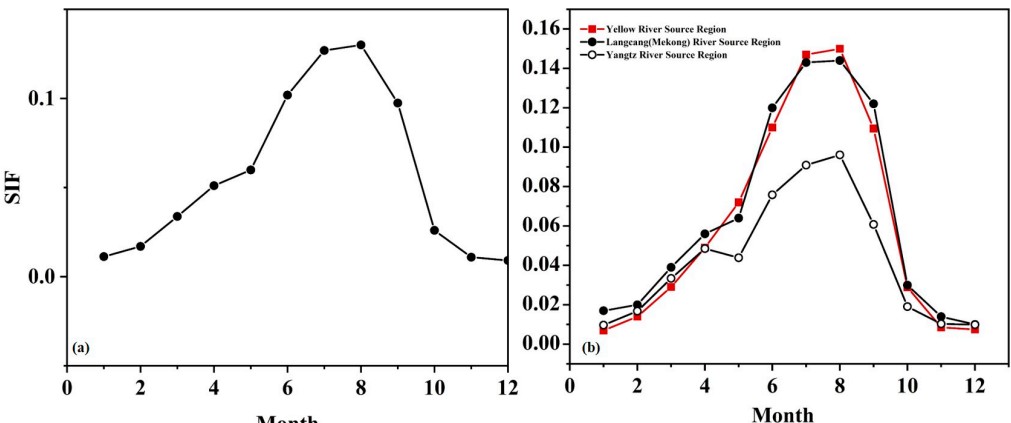

**Figure 5.** (**a**) Monthly variations in SIF in the TRHR; (**b**) monthly variations in SIF in YZR, LCR and YR.

### 3.3. Interannual and Monthly Variations in Meteorological Data in the TRHR

The variation rules of meteorological data are shown in Figures 6 and 7. The interannual changes in wind speed from 2001 to 2020 showed a trend of first increasing and then decreasing. The maximum value occurred in 2009, which was 2.33 m/s, and the minimum value occurred in 2013, which was 2.11 m/s. The maximum annual precipitation in the study area was 164 mm in 2018, and the minimum was 109 mm in 2002. From 2001 to 2020, the annual precipitation amount remained between 100 mm and 170 mm, showing a fluctuating trend. The highest annual mean temperature was 1.08 °C in 2016, and the lowest was −0.11 °C in 2001. The overall trend showed a gradual increase. The maximum relative humidity was 61.57% in 2019, and the minimum value was 51.40% in 2010, showing a trend of first decreasing and then increasing.

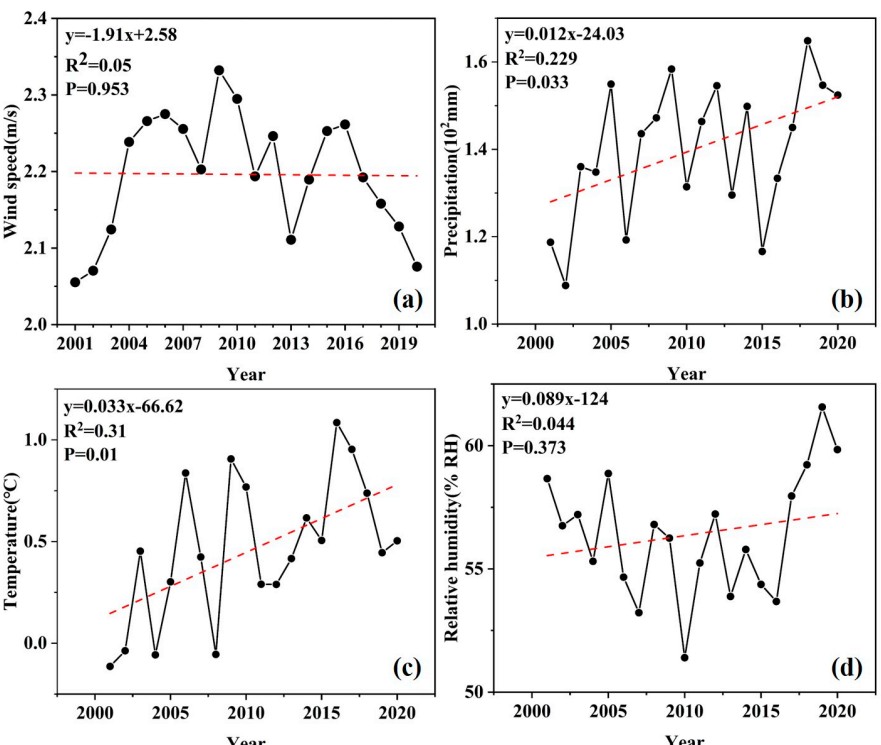

**Figure 6.** Changes in meteorological factors in the TRHR from 2001 to 2022. (**a**) denotes the interannual variation of wind speed; (**b**) denotes the interannual variation of precipitation; (**c**) denotes the interannual variation of temperature; and (**d**) denotes the interannual variation of relative humidity.

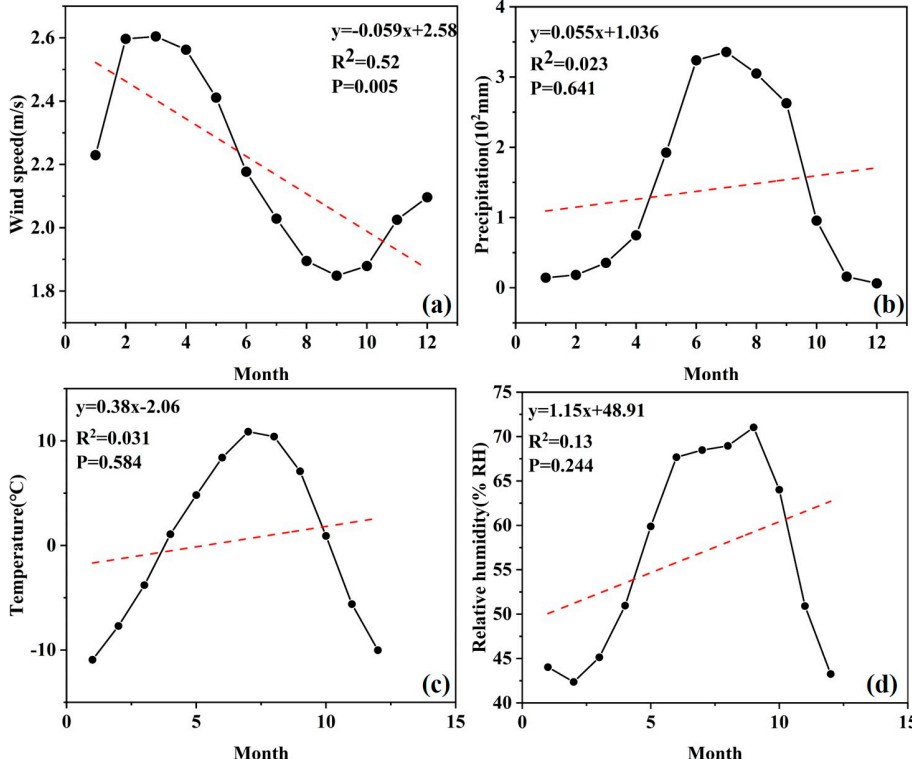

**Figure 7.** Changes in meteorological factors in the TRHR from January to December. (**a**) denotes the monthly variation of wind speed; (**b**) denotes the monthly variation of precipitation; (**c**) denotes the monthly variation of temperature; and (**d**) denotes the monthly variation of relative humidity.

The maximum monthly mean wind speed was 2.60 m/s in March and then began to show a downwards trend, continuing to decrease to 1.85 m/s in September and then increasing from October to December. The precipitation gradually increased beginning in January, reaching a maximum value of 336 mm in July, and then began to show a downwards trend. The minimum precipitation value was 6 mm in December. The monthly variation in temperature was consistent with the variation in precipitation; the highest temperature value was 10.89° in July, and the lowest value was −10.94°. The maximum relative humidity value was 71.03% in September, and the minimum value was 42.37% in February. The overall humidity change trend was highly consistent with those of both precipitation and temperature.

Based on the above research, the SIF in TRHR shows a gradual upward trend from 2001 to 2020, while the SIF in YR, LCR and YZR has a constant changing trend. The SIF values in LCR and YR were significantly greater than those in YZR. From January to August, the SIF value in TRHR showed an upward trend, reaching the maximum in August, and gradually decreasing from September to December. From May to October, the SIF value of YR and LCR was higher than that of YZR. This finding helps us to understand the change law of grassland, forest and other vegetation in TRHR, and provides a basis for studying photosynthesis and GPP, and also provides a reference for vegetation protection in TRHR.

## 4. Discussion

### 4.1. Feasibility Analysis of SIF Data

The SIF data used in this paper were calculated based on the TROPOspheric Monitoring Instrument (TROPOMI) on board Copernicus Sentinel-5P, as cited in the study of Chen (2022) [20]. In this study, Chen (2022) used machine learning to reconstruct TROPOMI SIF (RTSIF) values over the 2001–2020 period under clear-sky conditions at high spatial and temporal resolutions (0.05° and 8 days, respectively). The selected machine learning model achieved high accuracies on the training and testing datasets. The dataset was validated against the TROPOMI SIF and tower-based SIF values, and the results were compared with other satellite-derived SIF values (GOME-2 SIF and OCO-2 SIF). We anticipate that this new dataset will be valuable in assessing long-term terrestrial photosynthesis and constraining the global carbon budget and associated water fluxes. From our analysis of the SIF distribution map in the TRHR, it can be seen that the data can effectively represent the spatiotemporal changes in SIF in the TRHR and can also reflect the detailed changes in some small areas. The results show that the resolution of 0.05° is suitable for studying SIF changes in TRHR. In addition, the temporal resolution of eight days is also suitable for long-time-series change analyses. Due to the wide scope of this study area, this data can also be applied to other large-scale spatiotemporal changes of SIF. The spatial resolution of this data is not enough for small areas with high spatial resolution requirements.

### 4.2. Analysis of Climate Factors Influencing the Interannual Variations in SIF

Climatic factors are usually believed to be a crucial biophysical element affecting vegetation growth. By averaging the meteorological factors recorded at 18 stations in the TRHR according to the SIF time scale, the response relationships between the SIF values and meteorological factors were analysed. Precipitation and air temperature have been widely used to analyze the influencing factors of vegetation growth [26,27], and wind speed and relative humidity have also been selected to analyze the influencing factors of SIF in the TRHR. As shown in Figure 8, the wind speed and relative humidity have little effect on the annual variation in SIF in the TRHR, and the correlations are weak. There is a weak positive correlation between temperature and SIF with $R^2$ was 0.27. The correlation between precipitation and SIF is strong, with $R^2$ was 0.58. Other studies have also shown that the inter-annual vegetation change in the whole TRHR area has a strong positive correlation with temperature and precipitation [26,28,29]. Precipitation was the dominant factor affecting vegetation growth in the TRHR. Mainly because TRHR was an arid and semi-arid system, soil water availability in the surface soil layer is typically

more affected by rainfall events and evaporation than that in the deeper soils [30,31]. The mean soil volumetric water content in the surface layer (0–10 cm) is significantly correlated with precipitation, while that in the deeper layer (10–50 cm) is not significantly affected by rainfall in semiarid grasslands [32]. Plants with shallow roots that absorb water from the surface soil layer are more sensitive to soil water availability than those with deep roots, and they are more affected by precipitation. In contrast, deep-rooted plants rarely experience water stress due to their ability to draw on deep water reserves; thus, they respond less extensively and more slowly to rainfall events [33,34].

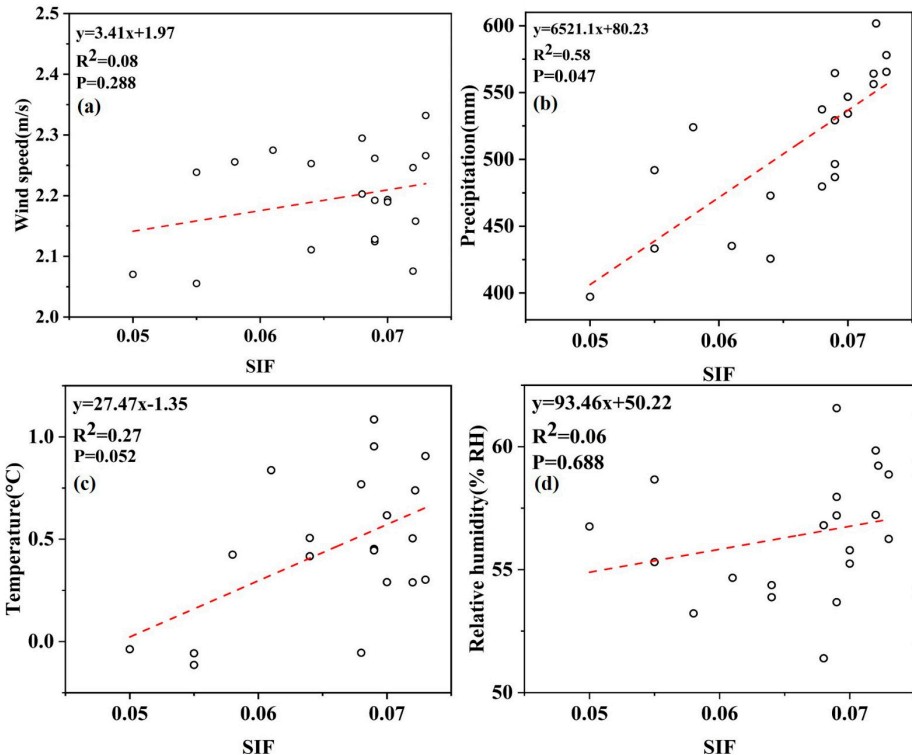

**Figure 8.** Relationships between annual average SIF and climate conditions from 2001 to 2020. (**a**) denotes the correlation between the annual average SIF and wind; (**b**) denotes the correlation between the annual average SIF and precipitation; (**c**) denotes the correlation between the annual average SIF and temperature; and (**d**) denotes the correlation between the annual average SIF and relative humidity.

In addition, some studies have shown that the linear responses of vegetation to meteorological factors such as temperature and precipitation in the TRHR are closely related to altitude [25]. The growth season had the most obvious change trend with altitude, and the change trend for the growth season was the weakest. The regularity of vegetation phenological change with altitude was not obvious when the altitude was below 3600–3700 m [25]. Moreover, the eastern part of the area has good hydrothermal conditions and a large range of wide valley basins in the mountains. This area has gained a concentrated population distribution, so the impacts of human activities cannot be ignored. Altitude and human activities may reduce the correlation between SIF and meteorological factors. Therefore, the effects of altitude and human activities need to be further considered quantitatively in subsequent studies [35]. At the same time, an average value of the whole region was taken every year, thus reducing the differences among regions, so the relationships between SIF and meteorological factors were low in the whole TRHR.

Furthermore, the 18 meteorological stations were divided regionally (YZR, LCR and YR), and the mean regional values were calculated; then, the response relationships between the SIF values in different source regions and the meteorological factors were analysed. As shown in Figures 9–11, the inter-annual SIF variations in YZR and LCR were closely related

to precipitation, but weakly related to other meteorological factors. This shows that the inter-annual variation of SIF in these two regions was mainly affected by precipitation. The inter-annual SIF variations in YR were significantly related to precipitation and temperature. With the increase in precipitation and temperature, SIF increases. The altitude of YZR and LCR is significantly higher than that of YR. The growth of vegetation in high-altitude areas is more susceptible to the effects of climate change, and its habitats are more fragile. Under the same climate change conditions, vegetation in low-altitude areas will be less affected, and the SIF value will be larger. Therefore, the factors affecting SIF in YZR and LCR are different from those in YR. Moreover, there are also differences in human activities within the three regions. Therefore, it is necessary to further analyze the influencing factors of SIF in the three regions in combination with altitude and human activities in future research.

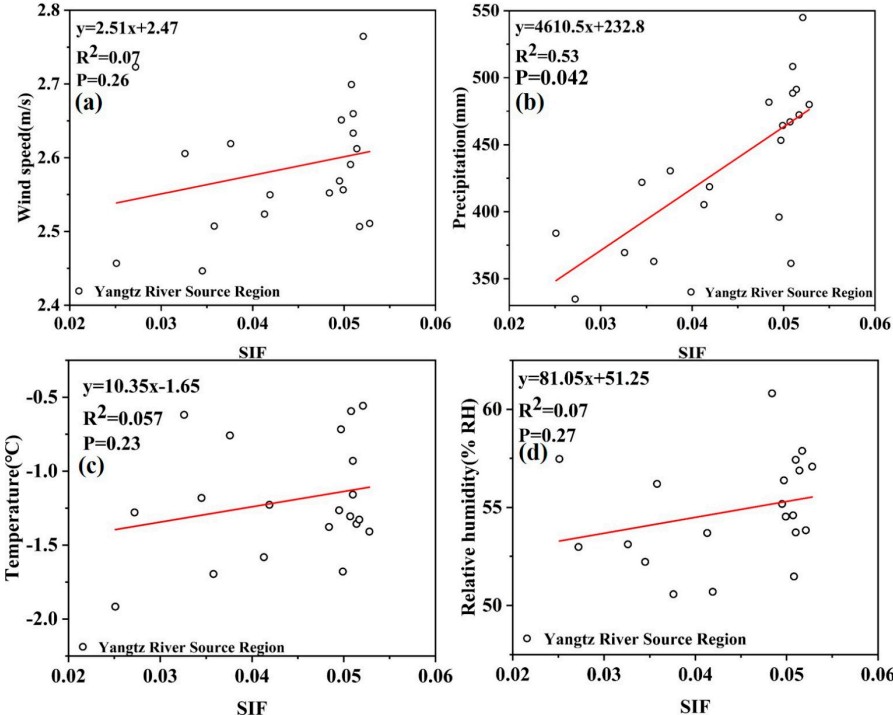

**Figure 9.** Relationships between SIF and climate conditions from 2001 to 2020 in YZR. (**a**) denotes the correlation between SIF and the mean annual wind; (**b**) denotes the correlation between the SIF and the mean annual precipitation; (**c**) denotes the correlation between SIF and the mean annual temperature; and (**d**) denotes the correlation between SIF and the mean annual relative humidity.

### 4.3. Analysis of Factors Influencing the Seasonal SIF Variation

Wind, precipitation, temperature and relative humidity were selected to analyze the factors influencing the monthly SIF changes, and the results are shown in Figure 12. SIF showed a positive correlation with precipitation in the 12th month, indicating that SIF showed an increasing trend with increasing precipitation. The correlation coefficient between SIF and precipitation was 0.81, indicating that the influence of precipitation greatly impacted SIF in this month. The correlation between temperature and SIF was 0.72, indicating that temperature also had a significant influence on SIF in all months. Moreover, relative humidity also had a high correlation with the SIF value, with an $R^2$ value of 0.56, indicating that the higher the humidity, the more suitable the growth of vegetation. The correlation between wind and SIF was weak overall, at 0.13, showing a negative correlation and indicating that the wind speed had a weak effect on vegetation growth. Therefore, precipitation, temperature and relatively humidity all impact the monthly variations in SIF in the TRHR. Different from the inter-annual variation of meteorological factors, the inter-monthly variation of meteorological factors is greater, especially precipitation and temperature. Precipitation is closely correlated with vegetation diversity and quantity

in high altitude areas [36,37]. Because of the interaction between mean precipitation and cumulative temperature (>10 °C) and mean temperature, the influence of annual mean precipitation on NDVI is significantly enhanced [38]. Temperature variation also influences the growth and development of plants. Environmental temperatures above or below the bearable temperature range of plants will negatively impact plant growth and development. Due to the interaction of temperature with soil type, landform type, precipitation, vegetation type, elevation, humidity index, and cumulative temperature, the influence of temperature on SIF is significantly enhanced. Therefore, precipitation, temperature and humidity will all have a greater impact on the monthly changes in SIF.

TRHR was divided into the source area of YZR, the source area of YR and LCR in this study, and the regional relationships between the monthly SIF variation and meteorological factors in the different areas were further analysed. The obtained meteorological data were also classified into these three source areas according to the site locations. The results are shown in Figures 13–15. Among these correlations, the correlation coefficients ($R^2$) of precipitation and the SIF values were 0.88, 0.89 and 0.89. Precipitation thus had the greatest influence on the monthly SIF variations, followed by temperature, with $R^2$ values of 0.87, 0.82 and 0.88. The correlations between relative humidity and SIF were 0.63, 0.75 and 0.7, indicating that humidity also had a certain influence on the SIF values. There was basically no correlation between the wind speed and SIF, and the wind speed thus had little effect on the monthly SIF variations. The results showed that the monthly variations in SIF values in the three regions were more correlated with precipitation, air temperature and relative humidity after partitioning. Perhaps it is because the statistics of SIF and meteorological factors were divided into different areas, which makes them more targeted. The effect of altitude can be reduced. It also shows that there is an urgent need for SIF data with high spatial resolution to conduct more detailed research on SIF in TRHR.

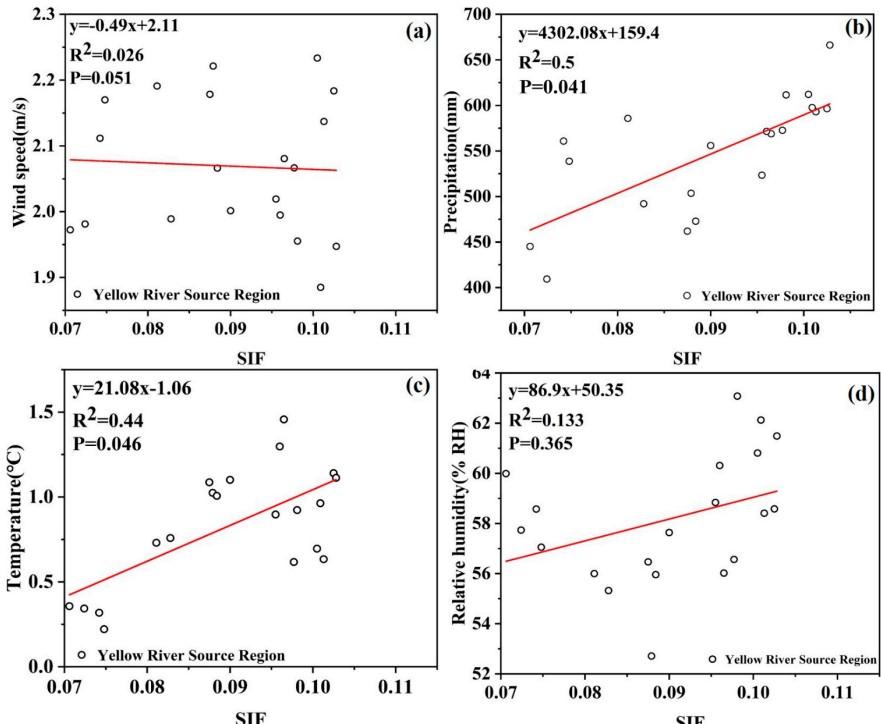

**Figure 10.** Relationships between SIF and climate conditions from 2001 to 2020 in YR. (**a**) denotes the correlation between SIF and the mean annual wind; (**b**) denotes the correlation between SIF and the mean annual precipitation; (**c**) denotes the correlation between SIF and the mean annual temperature; and (**d**) denotes the correlation between SIF and the mean annual relative humidity.

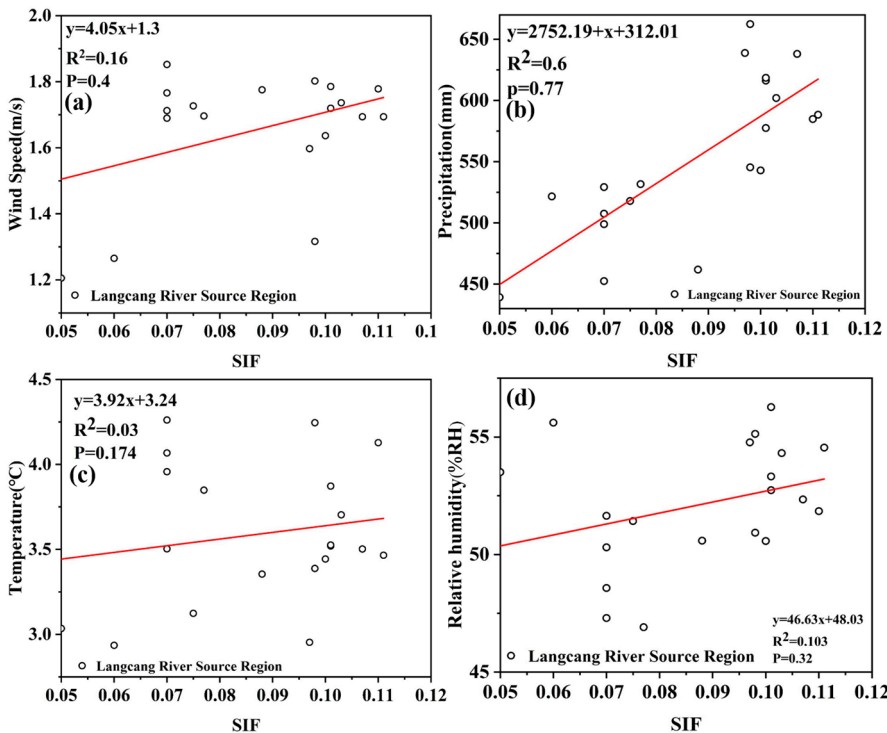

**Figure 11.** Relationships between SIF and climate conditions from 2001 to 2020 in LCR. (**a**) denotes the correlation between SIF and the mean annual wind; (**b**) denotes the correlation between SIF and the mean annual precipitation; (**c**) denotes the correlation between SIF and the mean annual temperature; and (**d**) denotes the correlation between SIF and the mean annual relative humidity.

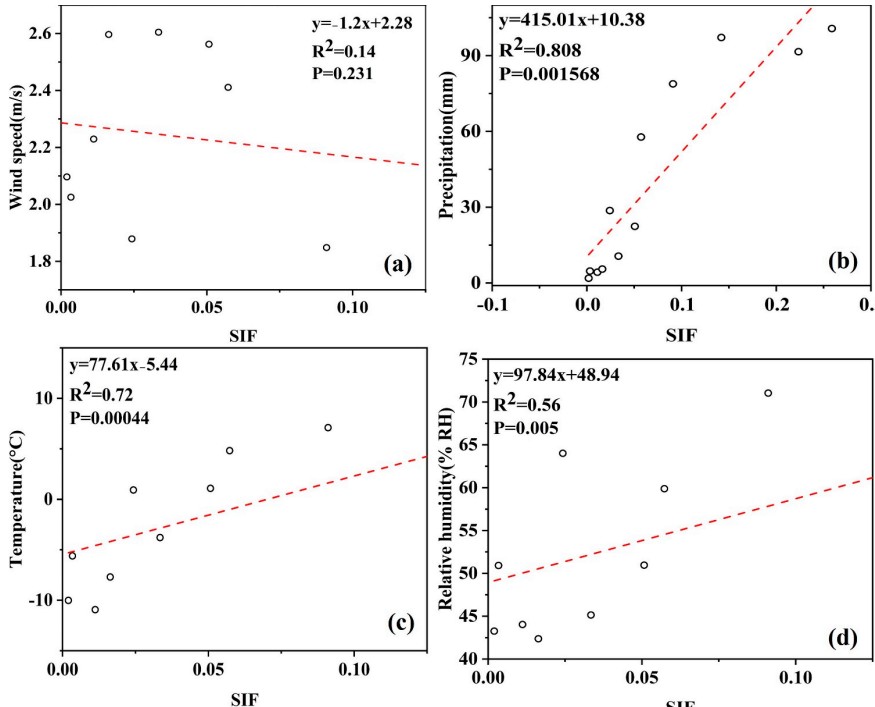

**Figure 12.** Relationships between SIF and climate conditions from January to December. (**a**) denotes the correlation between the mean monthly SIF and wind; (**b**) denotes the correlation between the mean monthly SIF and precipitation; (**c**) denotes the correlation between the mean monthly SIF and temperature; and (**d**) denotes the correlation between the mean monthly SIF and relative humidity.

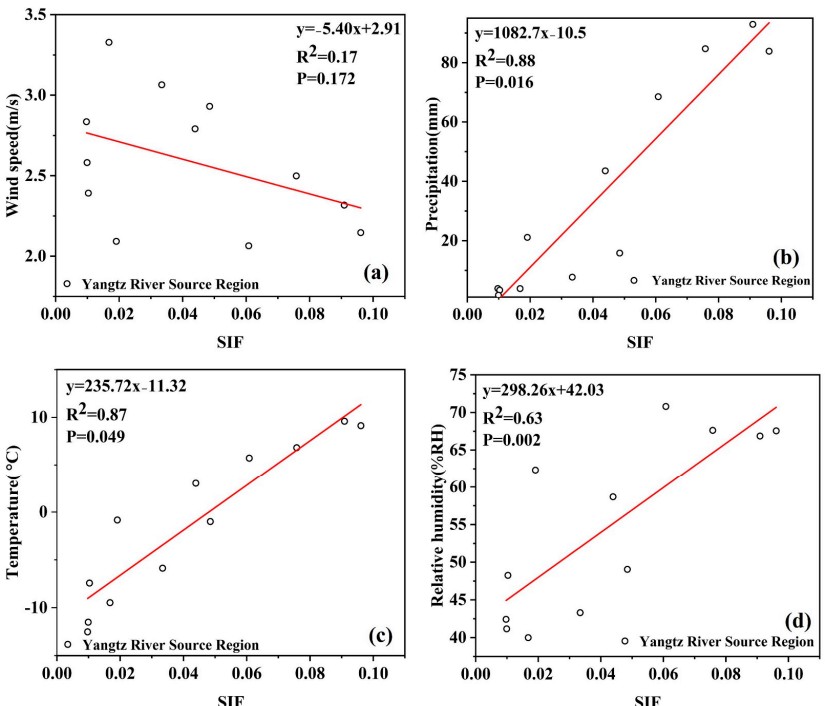

**Figure 13.** Relationships between SIF and climate conditions from January to December in YZR. (**a**) denotes the correlation between SIF and the mean monthly wind; (**b**) denotes the correlation between SIF and the mean monthly precipitation; (**c**) denotes the correlation between SIF and the mean monthly temperature; and (**d**) denotes the correlation between SIF and the mean monthly relative humidity.

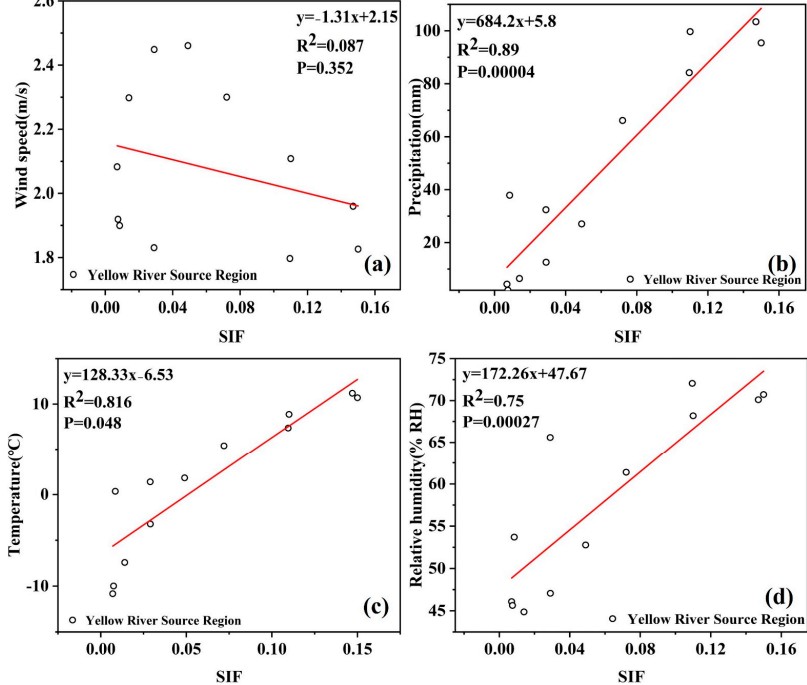

**Figure 14.** Relationships between SIF and climate conditions from January to December in YR. (**a**) denotes the correlation between SIF and the mean monthly wind; (**b**) denotes the correlation between SIF and the mean monthly precipitation; (**c**) denotes the correlation between SIF and the mean monthly temperature; and (**d**) denotes the correlation between SIF and the mean monthly relative humidity.

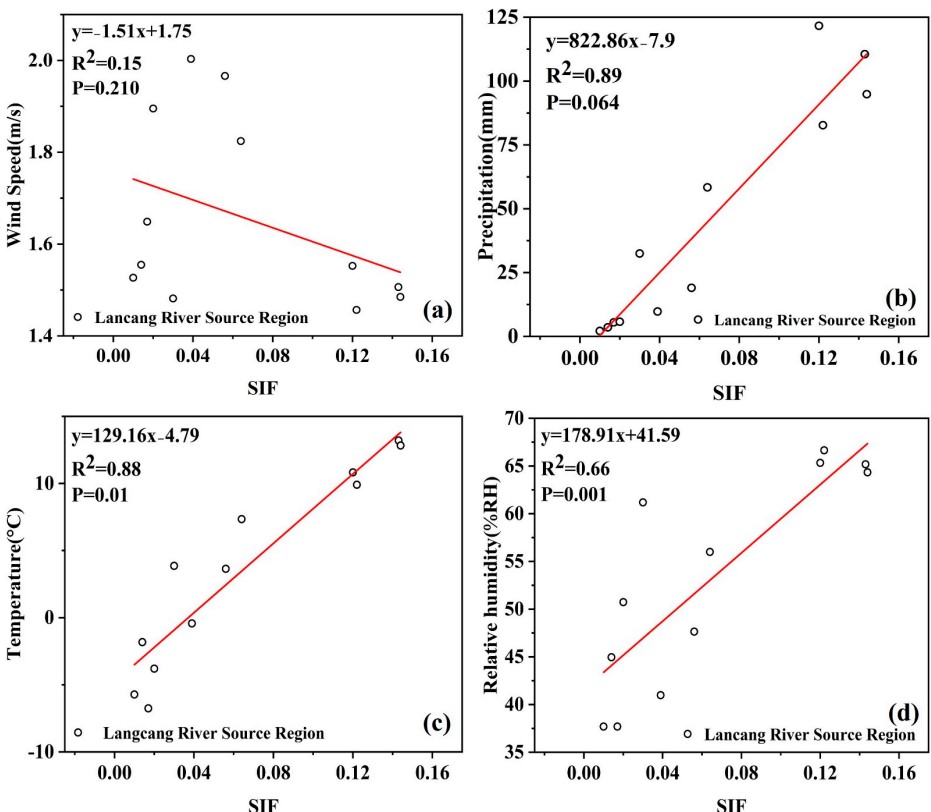

**Figure 15.** Relationships between SIF and climate conditions from January to December in LCR. (**a**) denotes the correlation between SIF and the mean monthly wind; (**b**) denotes the correlation between SIF and the mean monthly precipitation; (**c**) denotes the correlation between SIF and the mean monthly temperature; and (**d**) denotes the correlation between SIF and the mean monthly relative humidity.

### 4.4. Implications of SIF Monitoring and TRHR Environmental Protection

Vegetation plays the role of the largest carbon sink in the global carbon cycle. Understanding the dynamics of vegetation photosynthesis is an important goal that must be reached to achieve carbon neutrality. Remote sensing SIF is regarded as the most important breakthrough in estimating carbon uptake by remote sensing in recent years, and these data have been rapidly applied to large-scale vegetation photosynthesis calculations. Remote sensing technology can also contribute to a timely and comprehensive understanding of the occurrence, development, evolution, and migration process of problems in forest ecology. In the future, to realize refined monitoring and intelligent supervision, remote sensing monitoring will be strengthened, and advanced technologies will be used extensively, such as fixed-point continuous monitoring and rapid on-site monitoring. These advanced monitoring technologies should efficiently promote the sustainable development of the environment.

As the TRHR is an important ecological security barrier in China, it is critical to protect the ecology of the TRHR to ensure sustainable development. Benefiting from the efforts of the government to protect the environment, SIF in TRHR improved significantly over the past two decades. In the future, related management departments need to control and reduce total land-based pollution further and strengthen the supervision of the TRHR ecological environment. Human activities and social development should fully consider the impact on the TRHR environment, the quality of water environment, soil environment and atmospheric environment on the vegetation in TRHR. The results of this work can provide a reference for protecting the vegetation in the TRHR, which is of great significance for the ecological restoration and protection of this area. Despite the interesting results obtained, the SIF data adopted in this study has a low spatial resolution, so it is necessary

to further adopt SIF data with higher spatial resolution to conduct a more detailed analysis of the SIF in the TRHR. At the same time, the SIF changes and influencing factors in the TRHR should be comprehensively explored considering the impact of ground features such as lakes, combined with human activities, altitude and other factors.

## 5. Conclusions

For the first time, this study used the Global High-Resolution (8 days, 0.05°) Solar-Induced Fluorescence Dataset (2001–2020) to analyze the spatiotemporal variations of SIF in the TRHR. In addition, combined with meteorological factors, the influencing factors of SIF inter-annual and inter-monthly changes are discussed and analyzed. The main conclusions are as follows: (1) from 2001 to 2020, the SIF values in the TRHR fluctuated, ranging from 0.05 to 0.073, with a maximum value of 0.073 in 2005 and 2009 and a minimum value of 0.05 in 2002. The overall trend was gradually increasing. The SIF values in YR and LCR were significantly greater than those in YZR from 2001 to 2020; (2) the precipitation had greater effects on the inter-annual variations in the SIF values than other meteorological factors. With the increase of precipitation, the SIF value also showed an increasing trend; (3) regarding the seasonal variation, the SIF values of the TRHR in July, August and September were significantly higher than those in other months. The maximum value occurred in August at 0.11, and the minimum value was 0.008 in December. Beginning in May, the SIF values of YR and LCR increased significantly more than those of YZR, and these increases continued until September. (4) Precipitation, temperature and relative humidity greatly influenced the monthly variations in SIF, while wind speed had little effect.

In view of the low spatial resolution of the currently used SIF data, many details cannot be displayed, let alone detailed analysis for specific regions. In addition, the altitude difference in TRHR is relatively large, and the human activities in the eastern and western regions are greatly different. These factors should be considered. In the next step, SIF data with higher spatial resolution will be selected to further study SIF changes and factors such as elevation and human activities should be considered when analyzing changes in SIF.

The inter-annual and seasonal variations in SIF in the TRHR were analyzed in detail in this work, and the influence of meteorological factors on the change of SIF is analyzed; this research results are of great significance, as they provide references for vegetation protection in the TRHR. Meanwhile, the research methods and ideas in this paper can provide reference for the research into SIF in other areas.

**Author Contributions:** J.M. and R.A. conceived and designed the method and the experiments. J.M. conducted the field experiments and investigations. J.M. and F.X. performed data processing and analysis. J.M. wrote the original manuscript. J.M., R.A. and F.X. revised the manuscript. J.M. conducted the project administration and funding acquisition. All authors have read and agreed to the published version of the manuscript.

**Funding:** This research was founded by the National Natural Science Foundation of China (41871326).

**Data Availability Statement:** The data presented in this study are available on request from the corresponding author.

**Acknowledgments:** We sincerely thank the National Tibet Plateau Scientific Data Center for generously providing the SIF data as well as the China Meteorological Data Service Center for generously providing the meteorological data.

**Conflicts of Interest:** The authors declare no conflict of interest.

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
