# Peer review of "Characteristics of Solar-Induced Chlorophyll Fluorescence in the Three River Headwaters Region, Qinghai-Tibetan Plateau during 2001 to 2020"

_sustainability, doi:10.3390/su151914177_

Round 1

Reviewer 1 Report

This manuscript (sustainability-2558891) investigates the ecological health of the Three River Headwaters Region (TRHR) in China, specifically looking at the change in chlorophyll fluorescence, a key indicator of ecological well-being, from 2001 to 2020. The solar-induced chlorophyll fluorescence (SIF) values, which represent the level of photosynthetic activity in plants, ranged from 0.05 to 0.073, indicating variations in the region's ecology over time. This change was most pronounced in the source regions of the Yellow River and the Mekong River, with less change observed in the Yangtze River region. Seasonally, the SIF values were highest from July to September and lowest in December. It was also determined that climatic factors such as precipitation, temperature, and relative humidity significantly influenced these variations, as well as human activities and altitude. These findings can provide valuable insights for preserving the vegetation and overall ecology in the TRHR.

The introduction sections are good, but the results and discussion sections need modifications. The materials and methods section requires more detailed descriptions. The figures and images are good, but additional elements like scales, font sizes, etc., need to be modified for improvement. Regarding references, they should be added in the discussion section, demonstrating what is truly new and the contributions of this work. Why not compare with currently available models if this model is reported for the first time? However, overall, it is an interesting and well-written manuscript. However, it requires some modifications before it can be accepted.

Keywords in alphabetic order;

Please, the axis labels x and y are too small in Figures 1, 2 and 4. Additionally, all graphs need to include error bars, axis notes, and legends to ensure all information is easily accessible.

What value should I put in 'x' in Figures 5, 6 and 7 to use the equation?

The discussion section is a mixture of results and is not appropriate. The authors should rewrite it. Why are results presented in this section? What are the contributions of their models? Why were not the presented models discussed in the results section?

Why did not the authors include more recent bibliography, emphasizing the new findings with SIF modelling?

Check old references.

Grammar, spelling, and verbosity need to be checked.

Author Response

Dear reviewer,

Thank you very much for your email on 31 August 2023 with which you sent us the reviewer’s report on our paper (sustainability-2558891). We have carefully revised the manuscript according to reviewer’s suggestion. These opinions are of great help to the improvement of the paper. If the modification is not in place, please continue to guide us. Please see the attachment.

Author Response

(The authors gave the same response as above.)

Reviewer 3 Report

Review

Spatiotemporal patterns and influencing climate factors of solar-induced chlorophyll fluorescence in the Three River Head-waters region, Qinghai-Tibetan Plateau

Introduction provides concise and coherent background about the topic starting from the wider perspective (vegetation and their importance and stressors affecting them), followed by solar-induced chlorophyll fluorescence (SIF) as a tool to study photosynthesis and changes in carbon cycle, the link between SIF and remote sensing, and finishing with the objectives, which are in two folds 1) analysis of SIF in the Three River Head-wasters region, Tibetan Plateau and 2) investigating the effects of climate factors. The study utilized two sets of data (SIF, and metrological data) collected from sources in the public domain. The results section presents appropriately the main findings. This study provides an important account of utilizing publicly available data to monitor ecological environmental, and meteorological aspects with potential important application with remote sensing techniques.

However, I assume the discussion part could be restructured into a typical format in which findings are discussed. The current format of the discussion part sounds like a mix between methodology, and results and discussion. In this part, there might be a need to make it clear what is the purpose of the discussion. Is it to discuss the technicality in the method application, or is it to discuss the main findings in context with other relevant studies as well as explore the implications of the findings on the environment and the potentiality of adopting similar data to investigate environmental aspects?

It is good that the study reflects on the limitations in the conclusion as well as the potential future studies that could link the findings with anthropogenic activities.

Further suggestion: please check the resolution of Figure 1 (the coordinate numbers). Additionally, in the same figure, I could not distinguish distribution of meteorological stations (MS).

The sentence” ‘Although there are many lakes in TRHR region, it can be basically ignored on a large scale’ please provide more clarification (i.e., why it can be ignored?).

Abstract= ‘And’ at beginning of the sentence. Please check.  

The use of English is fine. Minor editing might be needed. 

Author Response

(The authors gave the same response as above.)

Reviewer 4 Report

I recommend “major revision”, in order to give the opportunity for research improving, considering the observations inserted in the file attached.

Author Response

(The authors gave the same response as above.)

Round 2

Reviewer 1 Report

I thank the authors for their responses. I still consider that the manuscript has been improved and is suitable for publication.

Minor grammar and spelling;

Reviewer 4 Report

All the required changes in the reviewing process were done and I agree with publication of the paper, from my side, in this form.